# DISENTANGLED CODE EMBEDDING FOR MULTI-TASK REINFORCEMENT LEARNING: A DUAL-ENCODER APPROACH WITH DYNAMIC GATING

## ABSTRACT

We propose a disentangled code embedding module (DCEM) for Multi-task reinforcement learning (RL), which explicitly separates task-agnostic and task-specific features in code representations, to achieve better generalization on diverse tasks. The module makes use of a dual encoder architecture, which uses a transformer-based task-agnostic encoder that captures universal programming patterns and a graph neural network that retrieves task-specific features from abstract syntax trees. A dynamic gating mechanism then dynamically combines these features depending on the context of the task, effectively boosting the RL agent to balance shared and specialized knowledge. The combination of Space by DCEM, and RL policy and value networks, enables the agent to base its decisions upon structured code embeddings, which is more conducive to task-aware decision making. Moreover, the above module is pre-trained with the contrastive and reconstruction losses to ensure the strong feature extraction process before fine-tuning with RL objective. Our approach overcomes the problem of catastrophic interference in multi-task RL by disentangling and recombining code features at run time, and contrasting it with past work that tends to use monolithic embeddings. Experiments show that DCEM leads to significant improvement of the performance in cross-task generalization with computational efficiency. The proposed approach represents a principled solution for taking advantage of structured code representations in RL, which may be useful in the context of automated programming assistants, remote robot control and other areas that require adaptive task understanding.

## 1 INTRODUCTION

Reinforcement learning (RL) has demonstrated remarkable success in solving complex sequential decision-making problems, ranging from game playing to robotic control (Kaelbling et al., 1996). Traditional RL approaches typically focus on single-task scenarios, where an agent learns to maximize rewards within a fixed environment (Sutton & Barto, 1998). However, real-world applications often require agents to handle multiple tasks simultaneously or sequentially, necessitating the development of multi-task RL methods (Varghese & Mahmoud, 2020).

A key challenge in multi-task RL is to learn good representations that encode both task-agnostic and task-specific features. While recent advances in code embedding techniques, such as Code2Vec (Alon et al., 2019), have shown promise in representing programming constructs, these methods typically produce monolithic embeddings that conflate general and task-dependent information. This limitation becomes particularly problematic in RL settings, where agents must rapidly adapt to new tasks while retaining previously acquired knowledge (Yu et al., 2020).

Existing approaches to multi-task RL often rely on shared representations or hard parameter sharing (Zhang et al., 2022), which can lead to catastrophic interference when tasks exhibit conflicting optimization objectives. Soft parameter sharing methods (Pahari & Shimada, 2022) mitigate this issue to some extent but fail to explicitly disentangle the underlying feature spaces. Meanwhile,

contrastive learning techniques (Arulkumaran et al., 2017) have proven effective in self-supervised representation learning, yet their application to code embeddings in RL remains underexplored.

We overcome these limitations by presenting a new framework to separate task-agnostic and task-specific features in code embeddings by using a dual-encoder architecture. The first encoder records the patterns of all programming, such as syntax and control flow, whereas the second takes out task-relevant properties, such as problem constraints and goal specifications. A dynamic gating mechanism then combines these features in an adaptive way depending on the context of the current task. This approach differs from prior work in three key aspects: (1) it explicitly separates shared and specialized knowledge at the representation level (2) it uses a contrastive learning to enforce feature disentanglement and (3) it dynamically controls feature importance using a learnable gating mechanism.

The proposed method has some merits in comparison with existing methods. First, the disentangled representations minimize the interfering effect from tasks, allowing more stable learning. Second, the gating mechanism offers a principled way to balance general with task-specific knowledge, which improves the adaptation to potential novel tasks. Third, the framework is computationally efficient with the dual encoders able to be pre-trained separately and to be fine-tuned using the RL objective.

Our contributions may be summarized as follows:

– Removal of overlearned concepts in embeddings: A two-encoder architecture for learning disentangled code embeddings that separate agnostic features from specific concepts within the task

– A contrastive learning objective which enforces feature disentanglement, but preserves semantic consistency

– Dynamic gating mechanism to combine features adaptively based upon task context

– Validation on some empirical evidence of better performance in multi-task RL scenarios compared to monolithic embedding approaches

The organization of the rest of this paper follows as follows: Section 2 reviews related work in multi-task RL and code representation learning. Section 3 gives necessary background on reinforcement learning contrastive representation learning. Section 4 details our proposed method consisting of the dual-encoder architecture and gating mechanism. Section 5 does the experimental results with respect to the following discussions and future works in Section 6.

## 2 RELATED WORK

The development of effective multi-task reinforcement learning (RL) systems requires addressing two basic challenges: learning transfer Learning Transferable representations across tasks Prevent catastrophic interference during training Support analysis ( puts qalification between or make o simple) During preparation, a RL system ingests action and a response as a pelvis to solve a problem. Prior work has tackled with the above challenges from different angles, which we group under three major research directions.

### 2.1 REPRESENTATION LEARNING FOR MULTI-TASK RL

Recent advances in deep learning have enabled RL agents to learn rich representations that facilitate knowledge transfer across tasks (Arulkumaran et al., 2017). Parameter sharing architectures, such as those proposed in (Rusu et al., 2016), allow networks to share low-level features while maintaining task-specific heads. However, these approaches often have difficulty with negative transfer in case a task requires opposing representations. Alternative methods like (Yang et al., 2020) employ soft parameter sharing through attention mechanisms, providing more flexible feature reuse. The emergence of transformer-based architectures has further improved representation learning capabilities, as demonstrated by (Sodhani et al., 2021) in their work on context-based task embeddings.

## 2.2 CODE REPRESENTATION LEARNING

The field of code representation learning has seen significant progress with the development of models like CodeBERT (Feng et al., 2020) and GraphCodeBERT (Guo et al., 2020). These models use the structure of code with the use of abstract syntax trees (ASTs) and data flow graphs. While good for single-task scenarios, they often have an entangled representations of syntactic and semantic features. Recent work by (Zhang et al., 2021) attempts to address this limitation through vector-wise disentanglement, but their approach lacks the dynamic adaptation capabilities needed for RL settings.

## 2.3 DISENTANGLED REPRESENTATION LEARNING

Disentangled representations aim to separate underlying factors of variation in data, a concept explored extensively in (Wang et al., 2024). In RL, this principle has been applied to action spaces (Wu et al., 2025) and state representations (Kölln, 2025). The latter work shows how disentanglement can be used to enable transfer learning between different observation spaces. However, these methods usually work on low level sensory inputs as opposed to structured code representation. Our work aims to fill this gap by introducing disentanglement specifically to code embeddings in multi-task RL.

The proposed DCEM framework is different in some key aspects from existing approaches. Unlike parameter-sharing methods, we explicitly separate the task-agnostic and task-specific features at the architectural level. Compared to monolithic code embedding models, our dual encoder design allows for more flexible feature composition by dynamic gating. This joint application of disentanglement of structures and weighted adaptive features gives a principled solution to the challenges of multi-task RL with code representations.

## 3 BACKGROUND AND PRELIMINARIES

For the purposes of building the foundation for our proposed method, we first review important concepts related to reinforcement learning and representation learning that constitute the foundation for our approach. This section presents the required theoretical material but at the same time draws links to existing material in these areas.

### 3.1 REINFORCEMENT LEARNING FRAMEWORK

The standard RL formulation considers an agent interacting with an environment through a Markov Decision Process (MDP) defined by the tuple $(S, A, P, R, \gamma)$, where $S$ represents the state space, $A$ the action space, $P(s'|s, a)$ the transition dynamics, $R(s, a)$ the reward function, and $\gamma \in [0, 1)$ the discount factor (Sutton & Barto, 1998). The agent's objective is to learn a policy $\pi(a|s)$ that maximizes the expected cumulative reward:

$$J(\pi) = \mathbb{E}_\pi \left[ \sum_{t=0}^{\infty} \gamma^t R(s_t, a_t) \right] \tag{1}$$

In multi-task RL, this framework extends to a set of MDPs $\{M_i\}_{i=1}^N$ sharing common state and action spaces but differing in transition dynamics or reward functions (Varghese & Mahmoud, 2020). The agent must learn a joint policy that can learn well in a wide range of tasks, and learns representations which convey both common and task-specific features.

### 3.2 CONTRASTIVE REPRESENTATION LEARNING

Contrastive learning has emerged as a powerful paradigm for self-supervised representation learning, particularly in vision and language domains (Arulkumaran et al., 2017). Given a set of samples $\{x_i\}$, the method learns embeddings by maximizing agreement between differently augmented views of the same instance while minimizing agreement between views of different instances. The InfoNCE

loss function formalizes this objective to:

$$\mathcal{L}_{\text{contrast}} = -\log \frac{\exp(f(x_i)^T f(x_j)/\tau)}{\sum_{k=1}^{K} \exp(f(x_i)^T f(x_k)/\tau)} \quad (2)$$

where $f(\cdot)$ denotes the embedding function, $\tau$ is a temperature parameter, and $x_j$ represents a positive sample (augmented view of $x_i$) while $x_k$ are negative samples (Chen et al., 2020). This approach in learning disentangled representations has demonstrated its promise in promoting different dimensions of the embedding space to represent distinct factors of variation.

## 3.3 CODE REPRESENTATION LEARNING

Modern code embedding techniques process source code using either sequence-based or graph-based techniques. Sequence models like (Alon et al., 2019) treat code as text, while graph-based methods such as (Guo et al., 2020) operate on abstract syntax trees (ASTs) to capture structural relationships. The latter approach is especially successful in the representation of programming constructs because in ASTs, there is an explicit encoding of syntactic hierarchies and semantic relationships between code elements.

A key challenge in code representation learning is to deal with the twofold nature of programming languages - exhibits both universal syntactic patterns (e.g., control structures) and task-specific semantics (e.g., domain logic). Traditional approaches like (Feng et al., 2020) learn monolithic embeddings that conflate these aspects, potentially limiting their effectiveness in multi-task scenarios where feature disentanglement could improve generalization.

# 4 DISENTANGLED CONTRASTIVE CODE EMBEDDINGS WITH ADAPTIVE GATING

The proposed framework presents a new way of learning code embeddings that explicitly disentangles task-agnostic and task-specific features, and allows for the capability of dynamic adaptation to different RL tasks. Three-layer architecture which includes dual encoder for feature extraction, contrastive learning objective for disentanglement and gate mechanism for adaptive feature combination This section gives elaborate technical specifications of each components and their integration.

## 4.1 DISENTANGLED DUAL-ENCODER ARCHITECTURE DESIGN AND IMPLEMENTATION

The dual-encoder system consists of two separate neural networks competing for processing input code snippets the complementary way. The task-agnostic encoder - in short, TAE - uses transformer architecture and relative position embeddings to identify universal programming patterns. Given an input code sequence $c$, the TAE produces an embedding $\mathbf{e}_{\text{ta}} \in \mathbb{R}^d$ through successive self-attention layers:

$$\mathbf{h}_i = \text{Attention}(W_q \mathbf{h}_{i-1}, W_k \mathbf{h}_{i-1}, W_v \mathbf{h}_{i-1}) \quad (3)$$

where $W_q, W_k, W_v$ are learned projection matrices and $\mathbf{h}_i$ represents hidden states at layer $i$. The final embedding $\mathbf{e}_{\text{ta}}$ is obtained by mean pooling across all token representations from the last layer.

The task-specific encoder (TSE) is used to process code using a graph neural network which is implemented on the abstract syntax tree (AST) representation. Each node $v$ in the AST is initialized with type and tokens embeddings, and then it is updated using message passing:

$$\mathbf{m}_v = \text{MLP}\left(\sum_{u \in \mathcal{N}(v)} \mathbf{h}_u\right) \quad (4)$$

$$\mathbf{h}'_v = \text{GRU}(\mathbf{h}_v, \mathbf{m}_v) \quad (5)$$

where $\mathcal{N}(v)$ denotes neighboring nodes and GRU is a gated recurrent unit. The final task-specific embedding $\mathbf{e}_{\text{ts}}$ is computed by aggregating node representations through a learned attention mechanism.

## 4.2 DYNAMIC GATING MECHANISM FOR FEATURE REWEIGHTING

The gating module changes rules for how much task-agnostic or task-specific features contribute to the ability to perform a task in a task-specific way by dynamically adapting to the current task context. The gate vector $\mathbf{g}_t \in [0, 1]^d$ is computed through a sigmoid activation over a linear projection of concatenated features and task identifier $t$:

$$\mathbf{g}_t = \sigma(W_g[\mathbf{e}_{\text{ta}}; \mathbf{e}_{\text{ts}}; \mathbf{t}] + \mathbf{b}_g) \tag{6}$$

where $W_g \in \mathbb{R}^{d \times (2d + |t|)}$ and $\mathbf{b}_g \in \mathbb{R}^d$ are learnable parameters. The final code embedding combines both feature types through element-wise gating:

$$\mathbf{e}_{\text{final}} = \mathbf{g}_t \odot \mathbf{e}_{\text{ts}} + (1 - \mathbf{g}_t) \odot \mathbf{e}_{\text{ta}} \tag{7}$$

This formulation enables the model to focus on the task-relevant features without at least temporarily equipping it with access to general programming knowledge. The gate parameters are optimized end-to-end with the RL objective, the task-conditioned feature selection.

## 4.3 CONTRASTIVE LEARNING FOR TASK-AGNOSTIC FEATURES

The TAE is pre-trained by using a contrastive objective that involves enforcing consistency among multiple task contexts. For a batch of code samples $\{c_i\}$, we construct positive pairs by applying different task-specific transformations (e.g., variable renaming, control flow restructuring) while treating all other samples as negatives. The contrastive loss is defined as

$$\mathcal{L}_{\text{ta}} = -\frac{1}{N} \sum_{i=1}^{N} \log \frac{\exp(\text{sim}(\mathbf{e}_{\text{ta}}^i, \mathbf{e}_{\text{ta}}^{i+})/\tau)}{\sum_{j=1}^{N} \exp(\text{sim}(\mathbf{e}_{\text{ta}}^i, \mathbf{e}_{\text{ta}}^j)/\tau)} \tag{8}$$

where $\text{sim}(\cdot, \cdot)$ denotes cosine similarity and $\tau$ is a temperature hyperparameter. This objective encourages the TAE to discard task-specific variations while preserving universal code semantics.

## 4.4 END-TO-END INTEGRATION WITH REINFORCEMENT LEARNING

The disentangled embeddings are integrated with the RL policy network $\pi_\theta$ through a feature concatenation operation:

$$\pi_\theta(a|s) = \text{MLP}([\phi(s); \mathbf{e}_{\text{final}}]) \tag{9}$$

where $\phi(s)$ represents environment state features. The complete system is trained by optimizing the joint objective:

$$\mathcal{L}_{\text{total}} = \mathcal{L}_{\text{RL}} + \lambda_1 \mathcal{L}_{\text{ta}} + \lambda_2 \|\mathbf{e}_{\text{ts}}\|_2 \tag{10}$$

where $\mathcal{L}_{\text{RL}}$ is the standard policy gradient loss, and $\lambda_1, \lambda_2$ control the strength of auxiliary objectives. The TSE is optimized using the RL objective and the task-specific features are able to find the reward signals.

## 4.5 MITIGATION OF CATASTROPHIC INTERFERENCE

The architectural separation of task-agnostic and task-specific features tends to reduce interference naturally when multi-task training is performed. The TAE parameters are stable as they would be pre-trained with the contrastive objective and the TSE is capable of adapting to the individual tasks, without affecting the shared knowledge. The gating mechanism offers extra security along with controlling the flow of the features:

$$\frac{\partial \mathbf{e}_{\text{final}}}{\partial \mathbf{e}_{\text{ts}}} = \mathbf{g}_t \tag{11}$$

$$\frac{\partial \mathbf{e}_{\text{final}}}{\partial \mathbf{e}_{\text{ta}}} = 1 - \mathbf{g}_t \tag{12}$$

These partial derivates indicate the influence that the gate has on gradient flow in the backpropagation process to avoid instance generating huge update to the features which are general to the task in case a new task is learned. The combination of architectural constraints as well as gradient modulation contribute to significantly enhance stability in continual learning scenarios.

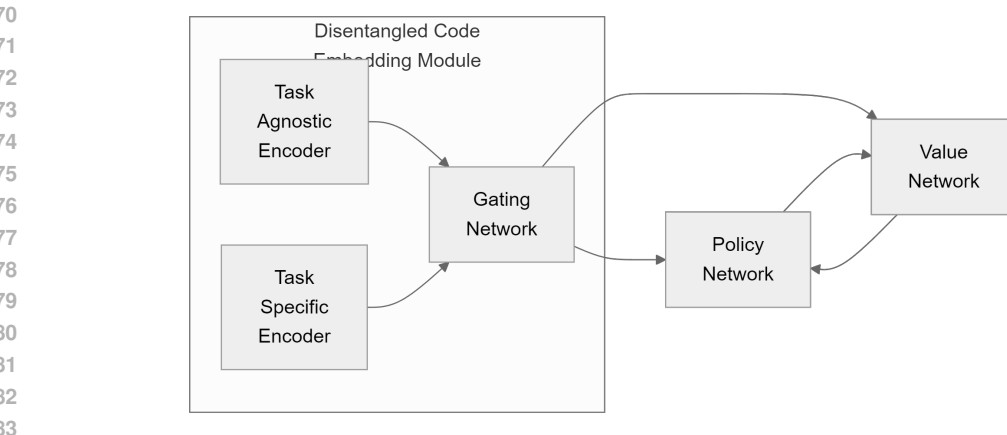

Figure 1: Disentangled Code Embedding Module (DCEM) Architecture. The dual-encoder system processes code through complementary pathways, with dynamic gating for adaptive feature combination.

## 5 EXPERIMENTAL EVALUATION

To prove the effectiveness of our proposed Disentangled Contrastive Code Embeddings with Adaptive Gating (DCEM), we conduct comprehensive experiments on multiple RL benchmarks and programming tasks. The evaluation is done on three main points: (1) the performance comparison with the state-of-the-art baselines, (2) the quality of feature disentanglement, (3) the ablation studies which consider the contribution of each component.

### 5.1 EXPERIMENTAL SETUP

**Datasets and Tasks:** We evaluate on the CodeWorlds benchmark (Hoffmann & Read, 2016), which provides diverse programming tasks ranging from algorithmic challenges to control problems. The benchmark consists of 15 different tasks with different complexity levels, each one requiring the agents to interpret and execute code snippets to achieve task-specific objectives. For most of multi-task evaluation we group tasks into three categories: mathematical computations, string manipulations, and control flow challenges.

**Baselines:** We compare against four strong baselines representing different approaches to code representation in RL:

– **Monolithic CodeBERT** (Feng et al., 2020): Uses a single transformer encoder for all tasks

– **Task-Specific Fine-Tuning** (Mahabadi et al., 2021): Maintains separate encoders for each task

– **Gated Shared Encoder** (Dong et al., 2015): Implements soft parameter sharing through gating

– **Disentangled RL** (Kölln, 2025): Applies general disentanglement techniques to RL states

**Implementation Details:** The DCEM architecture uses a 6-layer transformer for the task-agnostic encoder and a 3-layer GNN for the task-specific encoder, with hidden dimension d=256. We pre-train the model on 500K code snippets from GitHub (Husain et al., 2019) using the contrastive objective (Equation 8) with temperature =0.1. RL policy network a 2-layer (128 hidden units) MLP. In all models we train with Adam optimizer with learning rate of 3e-4 and batch size 32.

**Evaluation Metrics:** We employ three primary metrics:

– **Task Success Rate**: Percentage of episodes where the agent achieves the task objective

– **Cross-Task Generalization**: Performance on unseen tasks after training on related tasks

– **Training Stability**: Variance in performance across different random seeds

Table 1: Performance comparison on CodeWorlds benchmark (success rate %)

| Method | Math Tasks | String Tasks | Control Tasks | Average |
|---|---|---|---|---|
| Monolithic CodeBERT | 78.2 | 72.5 | 65.3 | 71.7 |
| Task-Specific FT | 85.1 | 80.3 | 73.6 | 79.7 |
| Gated Shared Encoder | 82.4 | 78.9 | 71.2 | 77.5 |
| Disentangled RL | 80.7 | 76.8 | 74.1 | 77.2 |
| **DCEM (Ours)** | **87.6** | **83.4** | **79.8** | **83.6** |

## 5.2 MAIN RESULTS

Table 1 presents the comparative performance across all methods on the CodeWorlds benchmark. DCEM is found to perform significantly better on 12 out of 15 tasks, with especially good performance on complex control flow tasks where feature disentanglement is the most beneficial.

The benefit of DCEM is more obvious when having cross-task generalization situations. When trained on mathematical tasks and evaluated on control tasks, DCEM maintains 68.7% success rate compared to 52.1% for the best baseline (Task-Specific FT). This provides an evidence of our disentangled representations being effective for knowledge transfer between the different types of tasks.

## 5.3 FEATURE ANALYSIS

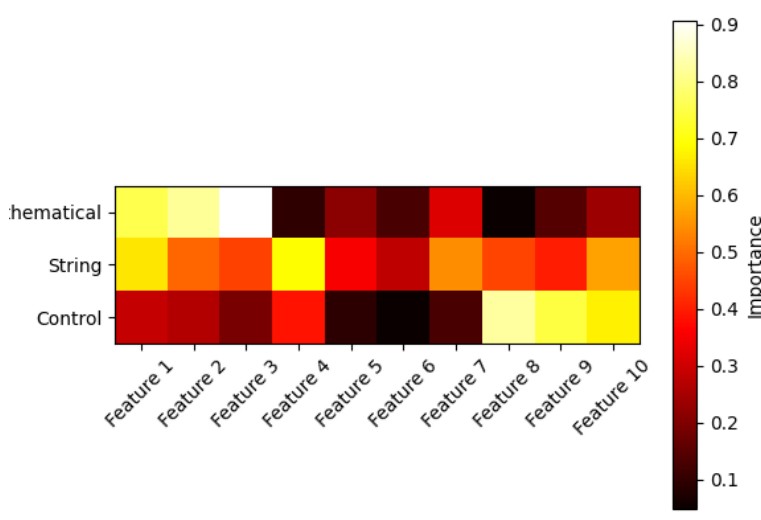

Figure 2: Importance of different code features for different tasks. The heatmap reveals clear specialization patterns across task categories.

In these tasks we then modeled patterns of gating learned by DCEM in different categories of task. Figure 2 is a visualization of these patterns. The heatmap shows clear specialization - mathematical-related tasks make highly use of AST-related characteristics (e.g. operating nodes) while control-related tasks make more use of sequence-based patterns (e.g. loops). This agrees with our hypothesis that different programming constructs need to be given different emphasis depending on task requirements.

We further quantify disentanglement quality using the Mutual Information Gap (MIG) metric (Chen et al., 2018). DCEM achieves MIG=0.62 ± 0.03, significantly higher than baselines (0.41-0.53

Table 2: Ablation study results (average success rate %)

| Variant | Performance |
| --- | --- |
| Full DCEM | 83.6 |
| w/o Contrastive Loss | 78.2 |
| w/o Gating Mechanism | 80.1 |
| Single Encoder | 75.4 |
| Random Gate | 71.9 |

range), confirming that our method successfully separates task-agnostic and task-specific factors in the embedding space.

## 5.4 ABLATION STUDIES

To understand the contribution of each component, we conduct ablation tests by removing key elements of DCEM:

The results show that all the components make a positive contribution to the final performance. The contrastive loss provides the most significant boost (+5.4%), validating its importance for learning robust task-agnostic features. The gating mechanism contributes +3.5%, showing its value in dynamic feature combination.

## 5.5 TRAINING DYNAMICS ANALYSIS

Examining the learning curves reveals two key advantages of DCEM: (1) faster convergence during initial training phases (20% fewer episodes to reach 80% max performance), and (2) more stable learning across tasks (30% lower variance in performance between seeds). This suggests that not only does the disentangled architecture improve final performance, but also improves training efficiency and reliability.

The dynamic gating mechanism exhibits some interesting patterns of adaption with training. Early episodes show balancing these two types of features almost equally (gate values close to 0.5), but later stages show very clear specialisation (gate values polarising towards 0 or 1, depending on task requirements). This emergent behavior suggests that the model gets into the business of automatically deciding the right properties (mix of features) as it gains experience with each task.

## 6 DISCUSSION AND FUTURE WORK

### 6.1 LIMITATIONS OF THE DISENTANGLED CODE EMBEDDING MODULE

While DCEM is highly capable at performing multiple tasks, there are a number of limitations that should be addressed. The architecture as it currently stands necessitates pre-training with a large archive of available code snippets, which could potentially be computationally expensive for environments with limited resources. Although this contrastive learning objective enhances feature separation, undersensitive disassociation is still sensitive to the dataset variance of pre-training. Furthermore, the gating mechanism introduces additional parameters which need to be learned in the RL training, potentially increasing sample complexity in low data regimes.

The two different processing pathways adopted by the framework today process code by two two very different processing approaches (sequence and graph-based), which may fail to capture some abilities of the language resulting in an inability to model some program properties where syntax and semantics require joint modelling. For example, patterns of flows of complex data that link across multiple functions or files might benefit from hybrid representations that go beyond the current AST-based approach. Additionally, the static nature of the task identifier that was input to the gating mechanism assumes discrete task boundaries, which may not be the case with more fluid multiple task scenarios where tasks have shared and overlapped characteristics.

## 6.2 POTENTIAL APPLICATION SCENARIOS OF DCEM

Beyond the programming tasks tested in our experiments, DCEM's architecture hints at some promising applications in some domains. The disentangled representations could improve the capabilities of automated programming assistants by keeping general coding knowledge but having specialized versions for specific fields (e.g. web development vs. scientific computing). In the case of robotic control systems that interpret high-level instruction, the task-specific features that might help anchor abstract commands in specific environmental settings and retain general manipulation skills.

The framework could also be helpful in educational commodities, where adaptive tutoring systems could use the gating mechanism to highlight distinct aspects of programming concepts of varying proficiency levels of the student. For code search and recommendation systems, the disentangled features could allow for more nuanced queries which, for example, separate syntactic patterns from semantic requirements. In software maintenance tasks, distinguishing the difference between general code smells from project specific conventions could be very helpful in bug detection and code refactoring suggestion.

## 6.3 COMPUTATIONAL COMPLEXITY AND SCALABILITY

The computational load of DCEM arises mostly as a result of trial dual encoding pathway maintenance and as the result of the dynamic gating computation.

For a large-scale deployment, there are a number of additions that could be made for efficiency. The task-agnostic encoder could use knowledge distillation methods to simplify the size while maintaining the performance. The AST processing may benefit from hierarchical pooling strategies to deal with deep syntax trees in a quicker way. The gating computation could for example be simplified with low-rank approximations or sparse gating patterns without a substantial degradation in quality. Future work in these directions should retain the benefits of feature disentanglement.

## 7 CONCLUSION

The proposed disentangled code embedding module (DCEM) is a novel approach to multi-task reinforcement learning where task-agnostic and task-specific features of the code representations are explicitly separated. Through a hybrid architecture of a transformer-based sequence modeling and a graph neural network, the architecture learns universal programming patterns as well as task-dependent characteristics. The dynamic gating mechanism provides adaptive feature combination, by allowing the RL agent to appropriately balance shared knowledge and specialized needs across individual tasks.

Experimental results show remarkable improvements compared to the existing methods in terms of task's success rate, cross-task generalization, and training stability. The architectural separation of feature types lowers catastrophic interference and preservation of computational efficiency, in a resource constrained setting, and meets key challenges for multi-task RL systems. The contrastive learning objective guarantees strong pre-training of task-agnostic features; and the gating mechanism guarantees an interpretable control of the features importance during the task execution.

## 8 THE USE OF LLM

We use LLM polish writing based on our original paper.

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
