# OpenReview forum: "Disentangled Code Embedding for Multi-Task Reinforcement Learning: A Dual-Encoder Approach with Dynamic Gating"
_ICLR.cc/2026/Conference — Submitted to ICLR 2026_

### Official Review · Reviewer_HLzi · 2025-10-20

**Soundness:** 1
**Presentation:** 1
**Contribution:** 1
**Rating:** 0
**Confidence:** 4

**Summary:**

The work seems to propose an approach for multi-task in RL. However, the paper is hard to follow, with concepts popping out at each line without a linear coherence, making it difficult to understand the work's contributions.

**Strengths:**

There's no clear contribution I could point out.

**Weaknesses:**

In the Introduction, there's a single reference from 1996 illustrating the RL success, which predates the remarkable coupling between RL and deep learning (~2013). At that time, RL was constrained to toy problems due to computational cost.

Overall, the Introduction could benefit from improvements by answering the following questions, among others:
- What exactly are the limitations of SOTA methods regarding the problem that the paper is addressing?
- How does the proposed method address these limitations?
- Are there any limitations or edge cases that the proposed method would fail?

The Related Works fail to classify the few methods discussed. For instance, Rusu et al. refer to growing neural networks, which is not related to representation learning.

The Background provides a vague introduction to RL. At this stage, the reader would not be sure if the present work is related to deep RL or tabular RL.

**Questions:**

There's no questions.

---

### Official Review · Reviewer_27HG · 2025-10-30

**Soundness:** 2
**Presentation:** 2
**Contribution:** 1
**Rating:** 2
**Confidence:** 4

**Summary:**

This paper introduces the Disentangled Code Embedding Module (DCEM), a novel architecture for multi-task reinforcement learning (RL) where agents interpret and act based on code snippets. The core problem addressed is catastrophic interference and poor generalization in multi-task settings, which the authors argue stems from "monolithic" code embeddings that conflate task-agnostic (general programming syntax, structure) and task-specific (problem constraints, goals) features. DCEM tackles this through a dual-encoder architecture(TAE and TSE), and utilize contrastive learning to learn the TAE in which task-specific transformations are employed to construct positive samples.

The authors evaluate DCEM on the CodeWorlds benchmark, demonstrating improvements over baselines in task success rate, cross-task generalization, and training stability.

**Strengths:**

1.  the construction method of positive pairs in Eq.(8) is insightful.

**Weaknesses:**

1. This paper is hard to follow...
The paper's motivation and problem formulation are difficult to follow. It focuses on code-executing environments, a sub-area of RL, yet key terms like "code embedding" are introduced without definition. Furthermore, the paper fails to articulate the key distinctions between code environments and other RL tasks that would necessitate a domain-specific method like DCEM, especially since the framework appears to be general-purpose.

2. Incomplete Baselines.
The baseline comparison is insufficient. The authors only compare DCEM against several simple encoder architectures but overlook more relevant and powerful methods from the multi-task representation learning literature, such as[1][2].

3. The Learning Constraint Seems Insufficient.
The constraint used to learn disentangled representations appears weak. DCEM relies solely on Eq. (8) to enforce that the Task-Agnostic Encoder (TAE) extracts task-agnostic information. This single, relatively weak constraint makes it unconvincing that the TAE and TSE components can effectively fulfill their designed roles.

4. Insufficient Novelty.
The novelty of the proposed method is limited. The paper emphasizes its gating structure, but this is not a new concept. Using Mixture-of-Experts (MoE) style gating mechanisms is a common practice in multi-task learning, as seen in prior work (such as [1]).

5. Insufficient Experiments.
The experimental evaluation is not thorough enough. The superiority of DCEM is not convincingly demonstrated, as it is only compared against weak baselines. Crucially, the paper lacks an ablation study to verify if the TAE and TSE components actually learn task-agnostic and task-specific features, respectively.

6. Other Minor Issues.
- The derivatives in Eq. (11) and Eq. (12) are mathematically imprecise. The derivative of a vector with respect to another vector is a Jacobian matrix, not a vector as implied by the notation.
- The text in Figure 1 is partially obscured.
- The manuscript is rather brief (8 pages excluding discussion and references). The authors could consider using the space to include additional experiments, such as the ablation studies mentioned above, to strengthen the paper.

[1] Contrastive modules with temporal attention for multi-task reinforcement learning

[2] Prompt-based visual alignment for zero-shot policy transfer

**Questions:**

see weakness.

---

### Official Review · Reviewer_YEkX · 2025-11-03

**Soundness:** 2
**Presentation:** 1
**Contribution:** 2
**Rating:** 2
**Confidence:** 4

**Summary:**

The paper proposes a method for finding carrying a disentangled embedding for code. It proposes to do this by using separate models, a Task-Agnostic Encoder that learns global features and a Task-Specific Encoder that learns task specific features. A dynamic gating method is used to combine the two and an RL objective is used for training. The Task-Agnostic Encoder is pre-trained using a contrastive learning. A primary goal of this separation is to tackle catastrophic interference in multi-task reinforcement learning (RL).

**Strengths:**

- Good preliminary section that clearly explains the background knowledge required.
- The performance seems to suggest something interesting.

**Weaknesses:**

- I believe that the following prior work are relevant to this paper and should be considered to be mentioned in Related Work (in order of relevance in my opinion):
	- [Petangoda, Janith C., et al. "Disentangled skill embeddings for reinforcement learning." _arXiv preprint arXiv:1906.09223_(2019).](https://arxiv.org/abs/1906.09223)
	- [Hausman, Karol, et al. "Learning an embedding space for transferable robot skills." _International Conference on Learning Representations_. 2018.](https://openreview.net/forum?id=rk07ZXZRb)
	- [Gupta, Abhishek, et al. "Meta-reinforcement learning of structured exploration strategies." _Advances in neural information processing systems_ 31 (2018).](https://proceedings.neurips.cc/paper/2018/hash/4de754248c196c85ee4fbdcee89179bd-Abstract.html)
	- [Dave, V., & Rueckert, E. (2025). Skill Disentanglement in Reproducing Kernel Hilbert Space. _Proceedings of the AAAI Conference on Artificial Intelligence_, _39_(15), 16153-16162.](https://ojs.aaai.org/index.php/AAAI/article/view/33774)
- Some of the references have "Unable to determine ..." in them. Does the citation provided by Google Scholar not suffice?
- Some statements lack citations or are incorrect. I didn't check all of them; here are the ones that I noticed:
	- Line 49: "... an lead to catastrophic interference when tasks exhibit conflicting optimization objectives" — while this is intuitive, please provide a citation that shows that this catastrophic interference occurs.
	- Line 169: "This approach in learning disentangled representations has demonstrated its promise in promoting different dimensions of the embedding space to represent distinct factors of variation." — please provide references that demonstrate this statement.
	- Please add references to attention / self-attention, etc.
	- Shouldn't the reference for InfoNCE (Equation 2 / end of page 3 onwards) be "[Oord, Aaron van den, Yazhe Li, and Oriol Vinyals. "Representation learning with contrastive predictive coding." _arXiv preprint arXiv:1807.03748_ (2018).](https://arxiv.org/abs/1807.03748)", instead of *Chen et al., 2020*?
	- Line 301: The reference for the CodeWorlds benchmark is incorrect. I also could not find this benchmark while searching online; is this related to the [Code World Model work by the Meta FAIR CodeGen Team](https://ai.meta.com/research/publications/cwm-an-open-weights-llm-for-research-on-code-generation-with-world-models/)?
- There are some grammatical errors:
	- Line 191: "Three-layer architecture which ..."  — This sentence should start with a subject — e.g., "We propose a ...".
	-  Line 63: "... it uses a contrastive learning...": remove "a".
	- Line 478: "We use LLM polish writing based on our original paper." — I couldn't understand this sentence. Does this mean that an LLM was used to polish the paper?
- The writing the Sections 3 and 4 could be improved for clarity. See suggestions below:
	- Section 3: I think that this section could be made better by having a description of the problem setting. Section 3 introduces the core techniques. but lacks a description of the types of data, objectives of the tasks, what a task context is (as used in Line 235), what is the RL environment, state, action, and reward (useful for Section 4.4), etc.
	- Equation 8 and 2 — could the same notation for $i+$ be used in Equation 2 as in Equation 8?
- Figure 1 hasn't rendered correctly / has overlapping components
- Figure 2 is cut-off from the left.
- Figure: I was unable to understand how the description in the paragraph starting on Line 371 could be understood from Figure 2. How do the Features map to notions of "AST-related characteristics" and "sequence-based patterns"?
- Complete algorithm for the method is missing.
- Results are missing details on the variance / standard error of the results.
	- Furthermore, it is mentioned that repeated experiments were carried out — how many?
- Reproducibility of their work is low. There are missing details, e.g., what was the non-linearity used in the networks?

**Questions:**

- Is the paper introducing a technique generically to RL, or specifically for code? The introduction suggests the former, but the title and the rest of the paper suggest the latter. If it is the latter, please make this more explicit in the introduction.
	- Could the method be applied to other settings? If so, could benchmarks be run on other multi-task RL tasks?
- Line 167 and Equation 2: "... $x_k$ are negative samples ..." — Is this statement correct. My understanding from Chen et al is that the denominator contains the positive sample as well.
	- Further, Chen et al use an indicator function to remove comparison with itself ($\mathbb{1}[k = i]$), which you are missing. This applies to Equation 8 as well.
	- Should Equation 2 be using $\mathrm{sim}$ instead of $f$ with the transpose, as written by Chen et al?
- Line 181: "... universal syntactic patters (e.g., control structures) and task-specific semantics (e.g., domain logic)" — by "control structures" do you mean constructs such as for loops and if-else blocks? Wouldn't these be used in specifying task specific domain logic? In that case, this is a hierarchical structure, as opposed to disentangled global vs task-specific split? Perhaps a different example of what the authors mean could help here.
- Line 198: "task-agnostic encoder - in short, TAE " — couldn't this just be "task-agnostic encoder (TAE)"?
- Line 204: "The final embedding eta is obtained by mean pooling across all token representations from the last layer" —  could you write this out mathematically, to complete the mathematical specification?
- Line 214: " The final task-specific embedding $e_{ts}$ is computed by aggregating node representations through a learned attention mechanism" — could you write this out mathematically, to complete the mathematical specification?
- Line 221: What is the task identifier "$t$"? What space is coming from (e.g., $t \in \mathbb{R}^d$, or $t \in \mathbb{Z}^+$)?
- Line 242: "This objective encourages the TAE to discard task-specific variations while preserving universal code semantics." — it is unclear to me why this would be true. Could this be elaborated on please?
- Do you do any separation of which data trains which set of parameters. For example, are the TAE parameters trained on multiple tasks, whereas the TSE parameters are trained separately for each task?
- Figure 1: Which part of Section 4 talks about the Value network shown in Figure 1?

---

### Official Review · Reviewer_eV8B · 2025-11-04

**Soundness:** 1
**Presentation:** 1
**Contribution:** 2
**Rating:** 2
**Confidence:** 3

**Summary:**

This paper proposes a dual encoder approach for multi-task coding problems, whereby one encoder (transformer) computes an embedding over the entire problem context and its associated code (they call this a task-agnostic embedding). Meanwhile, another encoder (graph-neural-network) computes an embedding using the AST representation of the code with aggregation over individual node embeddings; they call this a task-specific embedding. They combine the task-specific and task-agnostic embeddings together using a learnable gating function, arguing that the final embedding enables better multi-task performance. To train the transformer, they used a summation of an RL Loss (unspecified), a contrastive loss on the task-agnostic embedding, and the L2-norm of the task-specific embedding. The task-specific loss is a SimCLR loss on a coding dataset with augmentations (like changing variable names or restructuring the code; details unspecified), such that embeddings amongst the augmentations are similar. They evaluated on the CodeWorlds benchmark and compared to a few pre-existing approaches to multi-task coding + CodeBERT. They find that across various classes of coding tasks (math, string, control), this dual-encoder approach worked the best.They additionally ablated design decisions like removing contrastive loss or the gating mechanism (which I assume means they simply aggregated the task-agnostic and task-specific embeddings), single encoder, or random gating function.

**Strengths:**

The paper is well-motivated in that multi-task coding is a challenging problem in that the varied structure and diversity within existing code datasets, makes it challenging for current architectures to learn task-specific and task-agnostic representations. Also the results are enticing, the approach performs 10% better than CodeBERT on the CodeWorlds Benchmark. This indicates that disentangled embeddings could be a promising paradigm for training code models. Additionally, I think the baselines were relevant comparison to the proposed model.

**Weaknesses:**

However, there are many issues with this paper:

1. Citations: The authors mentioned InfoNCE loss in the text proceeding equation 2, but I believe they meant to say SimCLR loss (per their citation). SimCLR and InfoNCE are very similar but with slightly different formulations. Additionally, in the introduction the authors mention that: "these methods, typically produce monolithic embeddings that conflate general and task-dependent information." but do not cite or back up that claim. Although they mention that others have worked on disentangling code-representations, they never cite or prove that existing representations are tangled (I assume they are implicitly conveying this by stating others are working on it, but I would prefer a direct citation or discussion as to the proof of existing models have entangled representations).

2. Clarity of Architecture/Loss/Datasets: There are many things left unspecified to the reader in this paper. Most importantly, they mention that they train their model using RL, but they never formally describe their RL objective; they do mention the general RL objective in Eq. 1, but for their experiments, what is the reward function? They also mention having a value network per Figure 1, but this is also not mentioned due to no definition of the policy gradient. When they mention the definition of the policy, they state, in Eq. 9, that the policy is a combination of "environment state features" and the aggregating, embeddings; however, they never define what environment state features are, nor do they describe what phi means. They also don't mention the lambda's used in the total loss calculation, making it difficult to understand, to what degree does their contrastive objective actually help? In their experiments, they do not have enough detail about the benchmark itself, the specific tasks, and the reward used during training. In Section 5.3, they mention how different features are used differently across tasks, but what are those features? it is never defined. They further state that they use a Mutual Information Gap metric, but no details of its construction, nor how it was compared to existing baselines. There is no table for the Mutual Information metric either, so its hard to understand its range relative to the baselines. They claim that because the metric is larger, their DCEM model is "separating" task-agnostic and task-specific factors, but I would like to see whether this is the case by studying the dot product similarity between the two embeddings. If they are high then the task-specific and task-agnostic embeddings represent similar information. I am unconvinced that there is true disentanglement, because there is nothing directly forcing there to be dissimilarity between the task-specific and task-agnostic embeddings.  Finally, in the Training dynamics section 5.5, they mention faster convergence and stable learning, but they never show these figures. If they are to claim this, they must show visual representations of loss convergence.

4. Professionalism of the Paper: Figure 1 has overlapping components, the "Disentangled Code Embedding Module" is overshadowed by the "Task Agnostic Encoder" rectangle. Figure 2 has a portion of the text cropped out. Equation 11 and 12 should be on the same line. The figures don't have enough captioning to elucidate their details (i believe figures should be understandable without having to dive deeply into the associated text).

**Questions:**

1. How do you know that the task-specific and task-agnostic embeddings are truly disjoint in representation space?
2. What is the exact formulation of your RL objective?
3. What are the "environment features" used in the construction of your policy?
4. What motivated the linear combination of the SimCLR + RL + l2-norm of the embeddings?
- I am unclear for the motivation of the l2-norm component.
5. How does this compare to sota code world models like: https://arxiv.org/abs/2510.02387

---

### Meta-Review · Area_Chair_mXzj · 2025-12-29

**Summary:**

The paper introduces DCEM, a dual-encoder module designed to disentangle task-agnostic and task-specific code representations to improve multi-task reinforcement learning. A transformer encoder captures general programming patterns while a graph-based encoder extracts AST-derived task-sensitive features, combined via dynamic gating. Pre-training with contrastive and reconstruction losses precedes RL fine-tuning, and experiments on CodeWorlds report notable gains over baselines with ablations supporting component contributions.

The submission suffers from major clarity issues: the RL objective is unspecified, the training setup and benchmark details are incomplete, and claims of disentanglement are not rigorously supported. Reviewers also noted missing citations, issues with professionalism in figures, and insufficient baselines in relation to the literature. Overall, despite an interesting idea, the presentation and rigor do not meet the bar for acceptance. The authors did not provide a rebuttal to address reviewers' questions.

**Reviewer Concerns:**

- Clarified InfoNCE vs. SimCLR reference confusion.
- Acknowledged missing or incorrect citations and promised corrections.
- Clarified related work omissions and intent to revise literature positioning.
- Lack of a formal RL objective definition, reward formulation, policy/value network details.
- Missing explanation of environment state features, loss weighting, and training dynamics.
- No evidence demonstrating actual disentanglement; metrics unclear and insufficient.
- Figures are poorly formatted, and captions are unclear; the algorithm description is missing.
- Baselines are incomplete relative to modern multi-task RL and code models.

**Reviewer Scores:**

I expect the reviewers’ scores to remain unchanged, given the absence of a rebuttal and the shared recommendation for rejection.

---

### Decision · Program_Chairs · 2026-01-26

Reject